# Relevant Criteria for Improving Quality of Schizophrenia Spectrum Disorders Treatment: A Delphi Study

**DOI:** 10.3390/healthcare13222847

**Published:** 2025-11-10

**Authors:** Carlos Roncero, Alicia Sánchez-García, Llanos Conesa Burguet, Aurora Fernández Moreno, María Luisa Martin Barbero, Carlos Aguilera-Serrano, Verónica Olmo Dorado, Jon Guajardo Remacha, Joseba Rico Prieto, Clara Pérez-Esteve, Manuel Santiñá Vila, José Joaquín Mira Solves

**Affiliations:** 1Department of Health Sciences, Miguel de Cervantes European University, 47012 Valladolid, Spain; 2Psychiatry Unit, School of Medicine, University of Salamanca, 37008 Salamanca, Spain; 3Spanish Society of Dual Disorders, 28232 Madrid, Spain; 4Department of Health Psychology, Miguel Hernandez University, 03202 Elche, Spain; jose.mira@umh.es; 5Psychiatry Service of the General University Hospital of Valencia, 46014 Valencia, Spain; 6Spanish Society of Psychiatry and Mental Health, 28033 Madrid, Spain; 7Mental Health Working Group of the Spanish Society of Family and Community Medicine, 08009 Barcelona, Spain; 8Pharmacy Service, Gregorio Marañón Hospital, 28007 Madrid, Spain; 9Neuropsychiatry Group of the Spanish Society of Hospital Pharmacy, 28001 Madrid, Spain; 10Spanish Association of Mental Health Nursing, 28047 Madrid, Spain; 11Mental Health Group of the Spanish Society of Primary Care Physicians, 28009 Madrid, Spain; 12Spanish Society of Health Directors, 28020 Madrid, Spain; 13Spanish Association for Psychosis Support, 28027 Madrid, Spain; 14Atenea Research Group, FISABIO, 46020 Alicante, Spain; 15Spanish Society for Quality of Care, 33003 Oviedo, Spain; 16Institut d’Investigacions Biomèdica August Pi i Sunyer, 08036 Barcelona, Spain

**Keywords:** schizophrenia spectrum disorders, quality assurance, healthcare, patient safety, healthcare evaluation mechanisms, psychiatry

## Abstract

**Highlights:**

**What are the main findings?**
A Delphi study identified 26 quality criteria for schizophrenia spectrum disorder (SSD) care.Consensus prioritized key areas: early diagnosis, care coordination, and access.

**What are the implications of the main findings?**
Professionals and patients highlighted critical barriers in SSD healthcare.Results provide a foundation for a quality certification system in SSD care.

**Abstract:**

**Background/Objectives:** Schizophrenia Spectrum Disorder (SSD) represents a major challenge for healthcare systems due to its chronic nature, comorbid conditions, and high socioeconomic impact. Ensuring high-quality care for patients with SSD requires well-defined quality criteria based on consensus from healthcare professionals, patients, and caregivers. This study aims to identify and prioritize quality criteria for SSD care. **Methods**: A qualitative research approach was applied, including incorporating two focus groups—one with patients and caregivers (n = 7) and another with healthcare professionals (n = 8)—alongside the Delphi technique. The Delphi panel included 32 participants from psychiatry, primary care, mental health nursing, social work, and patient associations. The first round had an 88.9% response rate, while the second round achieved full participation (100%). The Delphi process was conducted and reported according to recommended guidelines for consensus methods (ACCORD checklist), specifying panel composition, rounds, predefined consensus thresholds, and controlled feedback between rounds. **Results**: A total of 26 quality criteria were ultimately selected, categorized into 16 identified barriers to effective care. Key priorities included early diagnosis protocols, coordinated multidisciplinary care, and improved access to specialized mental health services. **Conclusions**: The findings underscore the necessity of integrating patient experience into healthcare evaluation and highlight the potential for implementing a certification system to standardize SSD care across healthcare settings.

## 1. Introduction

The term Schizophrenia Spectrum Disorder (SSD) encompasses a group of disorders with heterogeneous etiology, characterized by significant variability in clinical manifestations, treatment response, disease progression, and prognosis [1]. These disorders include symptoms such as hallucinations, delusions, thought disturbances, lack of motivation, cognitive deficiencies, and reduced emotional expression, all of which profoundly impact the lives of affected individuals, their families, and their communities [2].

SSD is a chronic condition that requires continuous medical treatment [3]. Without proper management, individuals are at high risk of recurrent episodes, leading to a significant decline in health, cognitive impairment, well-being, and overall quality of life [4,5]. Affecting over 26 million people worldwide [6], SSD is associated with reduced life expectancy due to comorbid psychiatric disorders—such as depression, anxiety, and substance abuse [7]—as well as physical health conditions including metabolic syndrome, diabetes, cardiovascular and respiratory diseases, and infectious illnesses [8,9]. Additionally, SSD places a substantial burden on families and caregivers, contributing to significant socioeconomic costs, including unemployment, reliance on social assistance, and frequent hospitalizations during acute episodes [10]. Social stigma, which is commonly associated with SSD and other severe mental illnesses, along with reduced perceived social support, further exacerbates the psychosocial challenges faced by patients, negatively affecting their recovery and overall well-being [11].

SSD poses a considerable challenge to healthcare and social systems due to the need for continuous complex medical care, including pharmacological and psychosocial approaches [12], its functional impact on patients, and the frequent comorbidity with other mental and physical health conditions [13]. Long-term maintenance treatment and coordinated care have been associated with reduced physical morbidity and mortality in individuals with SSD, stressing the importance of sustained and integrated management approaches [14,15]. These aspects highlight the importance of ensuring specialized, comprehensive care based on well-defined quality criteria.

In this context, establishing and prioritizing quality criteria for SSD care is essential to improving treatment outcomes and enhancing patients’ quality of life. Identifying these criteria allows for the development of effective, patient-centered interventions that address the complexity of the disorder. Furthermore, optimizing care can help reduce the societal burden of SSD at both the healthcare and socioeconomic levels.

Recent studies have emphasized the need to develop specific quality indicators for mental health services, including schizophrenia care, to move beyond generic performance metrics [16,17]. These works underline the relevance of designing condition-specific, recovery-oriented, and patient-centered measures, which our study aims to address.

The aim of this study was to identify and reach a consensus on quality criteria for the care of patients with SSD, involving patients, caregivers, and professionals across all levels of healthcare. Through this process, the study aimed to lay the foundation for a quality certification system in SSD care, ensuring that healthcare services meet the highest standards of efficacy and equity, based on a broad and representative consensus. The purpose was to create a health quality model that serves as a reference for evaluating the quality of care provided by health centers to patients suffering from SSD. The consensus process was conducted at the national level in Spain, with the intended audience being mental health professionals, service managers, and policymakers seeking to improve quality standards in schizophrenia care.

## 2. Materials and Methods

To achieve the proposed objective of this study, qualitative and collaborative research methods were adopted, recognized for their effectiveness in data collection within the healthcare field [18,19]. Figure 1 illustrates the process of developing the quality criteria and the phases of the study.

Specifically, the focus group technique was employed, involving active participation from patients and professionals in the field. To reach a consensus on the identified quality criteria, the Delphi methodology was used, engaging a broader group that included both professionals and patients. As a preliminary step, an extensive literature review was conducted on the study topic, enabling the development of detailed scripts for the focus groups and the design of the initial questionnaire (Questionnaire 0), based on the most recent scientific evidence.

This study was conducted in strict adherence to the ethical principles outlined in the Declaration of Helsinki. The research received approval from the Ethics Committee of the San Juan de Alicante University Hospital on 29 November 2023, under reference number 23/074.

### 2.1. Literature Review

A comprehensive literature review was conducted to gather and analyze the most recent and relevant scientific evidence related to SSD and the quality criteria in their care. This review aimed to identify quality aspects in current healthcare models for individuals with SSD.

The databases used for this review included Medline (PubMed), ProQuest, and the International Journal of Integrated Care database, ensuring a comprehensive and high-quality selection of relevant studies. The findings from the literature review, conducted between May and June 2023, provided the foundation for developing the focus group guidelines. The insights gathered from these focus groups then informed the design of the initial Delphi questionnaire, ensuring that the study was structured based on both the most recent scientific evidence and the perspectives of key stakeholders.

This was a targeted literature review rather than a full systematic review, designed to support the development of study materials. Titles and abstracts were screened by two authors (A.S.-G. and J.J.M.S.), and recurring themes were extracted and grouped to inform the Delphi criteria.

The search strategy used for the literature review, including the databases consulted, search terms applied, and the number of results obtained, is available as Appendix A.

### 2.2. Patients and Informal Caregivers Focus Group

To gain a deeper understanding of the experiences and needs of individuals with SSD and their caregivers within the healthcare system, a focus group session was organized. This session included the participation of six patients previously diagnosed with SSD who were stable under treatment, as well as one informal caregiver directly involved in the daily care of a person with SSD (a total of three women and four men). Participants were recruited in collaboration with two SSD patient associations. These associations’ contribution to participant recruitment is detailed in Appendix A.

To facilitate interaction and encourage idea generation, a semi-structured discussion guide was used. This guide included questions aimed at exploring the challenges faced by patients and their caregivers throughout their interaction with various levels of the healthcare system, specifying different care levels such as primary care, emergency services, hospitals, mental health units, day hospitals, and other involved services (nursing, social work, etc.).

The session, which lasted a total of three hours with a mid-session break, was audio-recorded with the prior informed consent of all participants, ensuring their privacy and confidentiality. The recording was later transcribed verbatim for qualitative analysis. The session was moderated by two researchers (M.S.V. and A.S-G.) with experience in qualitative health research. A thematic content analysis was conducted to identify and group recurrent themes across participants, following the main principles of the COREQ framework.

### 2.3. Professionals Focus Group

Once the barriers in the care of individuals with Schizophrenia Spectrum Disorder (SSD) were identified through the focus group with patients and caregivers, a second focus group was conducted with professionals.

The objective of this session was to present and discuss the critical points previously identified and to reach a consensus on quality criteria to improve the care of these patients. Representatives from various associations and scientific societies specializing in the care of severe mental disorders were invited. Specifically, invitations were extended to associations in the fields of psychology and psychiatry, mental health nursing, primary care, hospital pharmacy, patient support organizations, and healthcare management. Of the 10 invited societies, 8 agreed to participate, each contributing a participant to the focus group. A detailed list of the scientific societies and professional associations involved in the professional focus group is provided in Appendix A.

The session was recorded with the prior consent of the participants, then analyzed qualitatively. Similar to the session with patients and caregivers, it was moderated by two researchers (M.S.V. and A.S-G.) experienced in qualitative health research, and a thematic content analysis was conducted to identify and group recurrent themes, following the main principles of the COREQ framework. As a result, a report was prepared summarizing the key findings and proposed improvements, which was reviewed and further refined by participants in an asynchronous process. This review formed the basis for the next phase of the study, in which the Delphi methodology was applied to prioritize and reach a consensus on the identified quality criteria.

### 2.4. Delphi

The Delphi method is a structured and iterative process used to achieve consensus among experts on complex topics, particularly in areas with limited empirical evidence or where professional judgment is essential. This methodology involves multiple rounds of surveys in which participants evaluate predefined statements or criteria, refining their responses in successive iterations until a stable consensus is reached. The Delphi technique is widely applied in healthcare research to establish quality standards, guidelines, and priorities for intervention [20,21]. Reporting of the Delphi process followed the ACCORD checklist to ensure transparency and completeness of methodological details.

Participants were recruited through professional associations and patient support organizations to ensure a multidisciplinary representation. A total of 36 individuals were invited, including healthcare professionals specializing in SSD care (psychiatrists, primary care physicians, mental health nurses, hospital pharmacists, and healthcare managers) and patients with firsthand experience of the disorder. Inclusion criteria included having at least three years of professional experience in the care of individuals with SSD (for clinicians) or being a diagnosed patient with substantial lived experience of the disorder (for users). Patient representatives were selected and invited by their respective associations to ensure independence and diversity of perspectives, while authors were not involved in the individual selection of participants. The panel size was defined to ensure multidisciplinary coverage (psychiatry, primary care, nursing, pharmacy, management, and patient representatives) and feasibility across rounds. The number of invited participants (N = 36) was set to secure at least 30 completers across two rounds, aligning with common practice in Delphi studies on health service quality. The panel achieved broad national representation, with participants from ten different Autonomous Communities of Spain, including Catalonia, Andalusia, the Basque Country, the Canary Islands, and others beyond the region where the study was conducted (Madrid). This distribution supports the generalizability of the findings within the Spanish health system. Each association and society was asked to nominate 2 to 4 representatives committed to completing at least two rounds of the Delphi process. An official email invitation was sent outlining the objectives of the study and requesting informed consent. Prior to the first round, the survey instrument was internally reviewed by the research team to ensure clarity and relevance, but was not externally piloted. Participation was voluntary, and panelists received a small reimbursement to compensate for time dedicated to the study, in accordance with ethical standards. Responses were collected anonymously using an online platform specialized in Delphi studies. Between rounds, aggregated results and mean scores for each item were shared with all panelists to allow them to reconsider their responses based on group feedback. The steering committee supervised the overall process but did not participate in scoring or decision-making. The Delphi was conducted in Spanish through a secure online platform, with plain-language explanations to ensure accessibility for patient participants. Templates of the focus group guide and Delphi questionnaire are available in Appendix A. Table 1 presents the sociodemographic and professional profiles of the final participants in the Delphi study.

The study was conducted in two rounds between November 2023 and March 2024. The initial list of 46 criteria was developed by combining the themes that emerged from the qualitative analysis of the focus group discussions with the main findings from the targeted literature review. This approach ensured that the items reflected both empirical evidence and stakeholder perspectives before the first Delphi round. In the first round, participants assessed 46 quality criteria identified in the previous research phases, rating their relevance on a 0-to-10 scale. Criteria that received a score of 9 or higher from at least 75% of respondents were accepted, while those rated below 60% were rejected. Items falling between these thresholds were re-evaluated in a second round, allowing participants to adjust their ratings based on group feedback. Consensus thresholds were predefined as ≥75% for acceptance and ≤60% for rejection, as summarized in Table 2.

## 3. Results

### 3.1. Literature Review

The literature review included a total of 174 studies published between 2013 and 2023, addressing four key thematic areas related to the quality of schizophrenia management: treatment adherence (65 studies), illness awareness (46 studies), integrated care (58 studies), and the patient journey (5 studies).

The findings highlight the importance of treatment adherence as an essential factor in preventing relapses, identifying barriers such as lack of illness awareness, complexity of the treatment regimen, and medication side effects. Effective strategies to improve adherence were identified, including psychoeducational interventions, family support, and the use of long-acting injectable antipsychotics.

Regarding illness awareness, results show its association with disease severity and treatment compliance, emphasizing promising interventions such as cognitive-behavioral therapy, transcranial stimulation, and self-observation through video recording.

On the other hand, integrated care in schizophrenia requires a multidimensional approach, stressing the need for early detection, personalized treatment, and the integration of pharmacological and psychosocial interventions to improve prognosis and patient quality of life.

Finally, the analysis of the patient journey revealed key factors influencing their trajectory within the healthcare system, highlighting the need to improve early diagnosis, reduce the duration of untreated psychosis, and strengthen coordination between healthcare and social services to promote social integration and patient recovery.

### 3.2. Focus Group

The focus group with patients allowed for the identification of critical points and the barriers they encounter in the care of individuals with SSD (Table A1, see Appendix B).

The consulted patient group also identified several positive aspects they had perceived throughout the treatment of their illness. The inclusion of the mental health nurse role stood out as an improvement in case management. Additionally, they appreciated positive advancements in the pharmacological field, with medications that have fewer and milder side effects and greater effectiveness in managing symptoms. They also noted improvements in inpatient facilities and day hospitals.

The participants in the healthcare professional focus group received the results obtained from the patient focus group, and an in-depth discussion was opened on the different aspects of each barrier, providing a variety of opinions based on their extensive experience and specific knowledge. Several difficulties and proposals were taken into account to improve care and diagnosis in severe mental disorders (SSD). Additionally, quality criteria were proposed to optimize these processes. Table A2 presents the quality criteria identified for each barrier as a result.

### 3.3. Delphi

The Delphi process began with the initial questionnaire (Questionnaire 0), which included 46 quality criteria. These were organized based on the 16 barriers identified during the focus groups, including two criteria that applied transversally to all identified barriers. Two rounds were conducted, achieving a participation rate of 88.9% (n = 32) in the first round and 100% (n = 32) in the second round.

To determine the acceptance or rejection of each quality criterion, the predefined cut-off points established by the promoting group were applied (Table 2).

This structured consensus process led to the final prioritization of 26 quality criteria, focusing on key aspects such as early detection, continuity of care, access to appropriate treatments, and integration of psychosocial support services. This set of criteria is detailed in Table A3.

## 4. Discussion

This study focuses on the main difficulties people with SSD face in receiving adequate care in the Spanish National Health System and establishes a list of quality criteria to guide the work of health services to address them.

For the first time, a proposal to improve the quality of care for people with SSD is addressed through clinical and patient consensus. This can add value to SSD care processes if taken into account and implemented, as it emphasizes critical points prioritized by clinicians and patients.

Recent research highlights persistent gaps in quality measurement for schizophrenia and serious mental illness. Recent reviews [16,17] have shown that most existing indicators remain process-focused and often fail to capture patient experience, recovery goals, and cross-level coordination. Similarly, initiatives such as the Italian *Patient Journey* project have reinforced the importance of developing user-centered criteria, reporting barriers comparable to those found in our study, such as insufficient communication, limited involvement in care plans, and lack of service integration [22]. These findings further support the relevance of our approach, which integrates both clinical and experiential dimensions of care. The identified barriers are based on what patients have expressed and are consistent with findings from other studies [22,23,24,25,26,27]. Delays in receiving care, communication issues, knowledge of the disease, and its stigmatization—especially in the case of mental health—along with the necessary social support [28], are the most significant factors hindering the proper care of individuals suffering from this condition.

The consensually defined quality criteria can help improve the care process for patients with SSD, as has been proposed in other pathologies [29]. These criteria serve as the foundation for the ongoing quality improvement efforts, enabling the establishment of an accreditation system for them [30].

In our study, rejected criteria mainly referred to educational, organizational, or advocacy-oriented actions—such as awareness campaigns, staff training, or formal coordination protocols—that were perceived as less feasible or less directly impactful in daily clinical practice. In contrast, items related to clinical management, care continuity, and patient–professional communication achieved a stronger consensus. This suggests that participants prioritized pragmatic and immediately applicable measures over broader structural or policy-level reforms. It is noteworthy that, as reported in recent international reviews, a similar pattern emerges: while system-level, advocacy, and educational actions are widely recognized as important, patients and clinicians often prioritize clinical, continuity, and communication measures for immediate practical impact. This convergence suggests that to produce real changes in the quality of care for SSD, consensus processes tend to favor actions that are rapidly applicable and directly perceived by end-users, without excluding the long-term value of broader structural reforms [16,17].

Significant progress has been made in the care of individuals with SSD, and the Mental Health Strategy of the National Health System for the period 2022–2026 [31], along with previous efforts [32], addresses some of the barriers identified in this study, outlining the necessary actions. In this context, the proposed quality criteria would be highly beneficial in guiding these efforts.

Recent studies also highlight the importance of interdisciplinary collaboration in improving the quality and continuity of mental healthcare, particularly during transitions of care. The contribution of clinical pharmacists has been shown to enhance medication reconciliation and safety for patients with SSD [33,34]. Integrating such collaborative practices supports the implementation of several of the proposed quality criteria in this study.

Improving care for individuals with SSD represents a multifaceted challenge that requires a comprehensive and coordinated intervention across multiple levels of healthcare. The implementation of 26 consensually defined quality criteria provides a pathway toward more effective, personalized, and humane care for these patients. By addressing critical barriers such as delays in diagnosis, insufficient specialized consultations, communication improvements, and enhanced training, early diagnosis and appropriate treatment can be facilitated. This would ultimately contribute to a better quality of life by minimizing the disease’s impact on daily life and reducing the social stigma associated with mental illness [11,24,25,28]. Future research could validate these quality criteria in broader or international contexts to assess their applicability across different healthcare systems and cultural settings.

This research is based on participants’ assessments, so the selection of indicators may be influenced by biases inherent to their individual opinions and experiences. However, professionals from different centers and geographic areas were invited to ensure a broader representation of perspectives and experiences, thereby strengthening the validity and applicability of the selected indicators. Nevertheless, as the study focused on Spanish experts and patient representatives, the generalizability of the findings to other health systems may be limited. Finally, although the qualitative analyses may also involve some degree of subjectivity, multidisciplinary participation helped minimize this bias.

## 5. Conclusions

Improving the quality of care for patients with schizophrenia spectrum disorders requires addressing the barriers that hinder effective care. Therefore, 26 quality criteria have been consensually defined and prioritized, whose implementation may help enhance the quality of care for individuals suffering from these conditions, such as schizophrenia. We propose that those criteria can be used to improve the quality of healthcare in people with SSD and enhance the management and organization of mental health services.

## Figures and Tables

**Figure 1 healthcare-13-02847-f001:**
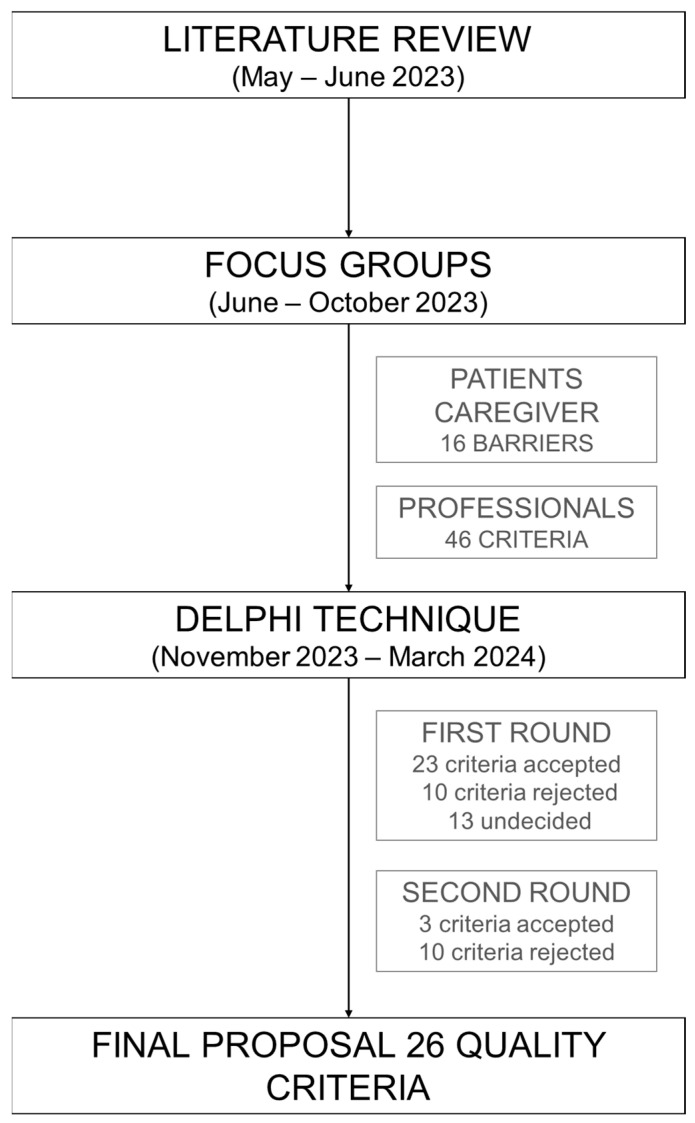
Development and phases of the study.

**Table 1 healthcare-13-02847-t001:** Sociodemographic and Professional Profile of Delphi Study Participants.

	N	%
**Sex**		
Women	23	71.88
Men	9	28.13
**Age**		
Mean age	49.41	
Standard deviation	10.17	
**Professional Profile**		
*Professionals:*		
Psychiatrist	11	34.38
Physician (non-psychiatrist)	7	21.88
Specialist Pharmacist in Hospital Pharmacy	4	12.50
Mental Health Specialist Nurse	4	12.50
Director	2	6.25
*Patients:*		
Janitor	1	3.13
IT specialist	1	3.13
Administrative Staff	1	3.13
Retired	1	3.13
**Place of origin**		
Community of Madrid	8	25.00
Catalonia	4	12.50
Andalusia	3	9.38
Aragon	3	9.38
Basque Country	3	9.38
Valencian Community	3	9.38
Principality of Asturias	2	6.25
Canary Islands	2	6.25
Galicia	2	6.25
Castile and Leon	1	3.13
Chartered Community of Navarre	1	3.13

**Table 2 healthcare-13-02847-t002:** Cut-off Points for Quality Criteria.

Round	Consensus Level	Rating ≥ 9
1st round	Acceptance	Agreement rate ≥ 75%
	Rejection	Agreement rate ≤ 60%
	Moves to the 2nd round	Agreement between 60% and 75%
2nd round	Acceptance	Agreement rate ≥ 75%
	Rejection	Does not meet acceptance criteria

## Data Availability

All data generated or analyzed during this study are included in this published article and its Appendix A.

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
