# Peer review of "Relevant Criteria for Improving Quality of Schizophrenia Spectrum Disorders Treatment: A Delphi Study"

_healthcare, 2025, doi:10.3390/healthcare13222847_

Round 1
Reviewer 1 Report
Comments and Suggestions for Authors
Relevant criteria for improving quality of Schizophrenia Spectrum Disorders treatment: a Delphi study
This study describes an important topic useful for daily practice. The paper is well written, but some limitations, including methods and discussion, should be described. I suggest a minor review.
Abstract: Please specify the guidelines followed for the Delphi technique
Keywords: Please check that all keywords are MESH keywords
INTRODUCTION
I will suggest that long-term treatment minimises mortality in patients with SCH. 20-year follow-up study of physical morbidity and mortality in relationship to antipsychotic treatment in a nationwide cohort of 62,250 patients with schizophrenia (FIN20). World Psychiatry. 2020 Feb;19(1):61-68.
MATERIAL AND METHODS
The literature review is unclear. It does not explain how it was done or who performed it. Did the authors use PRISMA or a non-systematic review?
Not clear how the authors defined the necessary number of participants in Deplhi.
Not clear how the authors defined the 46 criteria for Delphi.
Please give the “cut-off” points for this study.
RESULTS
Clear
DISCUSSION
Please expand the discussion in the context of other papers and limitations.
Authors could mention some studies focused on the collaboration between different specialists in mental health, such as clinical pharmacists; Curr Opin Psychiatry. 2022 Sep 1;35(5):332-337 & Clinical pharmacist interventions in the transition of care in a mental health hospital: case reports focused on the medication reconciliation process. Front Psychiatry. 2023 Dec 27;14:1263464. This is especially important at the transition of care, mentioned by the authors.
Reviewer 2 Report
Comments and Suggestions for Authors The study addresses Schizophrenia Spectrum Disorders (SSD) and aims to establish consensus-based criteria for quality of care. The paper is well-written and structured. The mixed-methods approach using focus groups with patients and professionals, and the Delphi consensus process is appropriate. High participation rates were achieved, indicating strong commitment from the panel and robust consensus. Few minor comments to be addressed by the authors:- What quality criteria are currently missing or inadequate for SSD?
- The authors need to update their literature review to include 2024-2025, this ensures the study's findings and proposed criteria are grounded in the complete latest evidence and context.
- The authors need to specify the method of qualitative analysis used for the transcripts of the audio-recordings. Clarifying the method for example: thematic analysis will enhance methodological accuracy.
- For the rejected criteria, the Delphi process was rigorous, rejecting a total of 20 criteria across two rounds (10 in the first, 10 in the second, and 3 that did not meet criteria from the undecided list). To enrich the discussion, it would be beneficial to briefly describe general characteristics of the criteria that were rejected this would provide valuable context for why the final 26 criteria were selected over others.
Reviewer 3 Report
Comments and Suggestions for Authors
Thank you for the opportunity to review this paper. It is an interesting and important study and I enjoyed reading it. However, the following issues need to be addressed before it is suitable for publication:
- Introduction
- Provides useful clinical background on schizophrenia care and rationale for the current study.
- However, there is no indication of current research on quality care indicators, either for schizophrenia specifically or mental health conditions generally. Please add this to better orient the reader.
- When explaining your aims, please more clearly state your intended audience for the consensus exercise and the geographical scope (national, regional or global)
2. Methods
- Methods used are appropriate to address the research aims but more details is needed when reporting these.
- For the literature review, please ensure that reporting is in accordance with the relevant PRISMA checklist (or similar). Some details such as how information from the literature review results are aggregated is missing.
- Similarly, your methods for the focus group component are relatively brief and information such as the qualitative approach used to analyse focus group data (e.g. thematic or content analysis) and who conducted the focus groups is missing. Use of a reporting checklist for qualitative research such as COREQ can help to ensure that all relevant details are recorded.
- Use of a reporting checklist such as ACCORD for the Delphi would also be useful here to ensure all relevant information is included.
- Please provide more information on inclusion criteria for panellists- was it just that they were a member of a group that you outlined or were other criteria applied?
- Who was responsible for panellist selection?
- Please state whether panellists were able to or required to explain their responses and whether they could propose new items.
- Please outline any piloting of the study materials and/or survey instruments and changes made as a result of this.
- Please describe how feedback was provided to panellists at the end of each consensus step.
- Please state if the steering committee was involved in decisions made by the consensus panel.
- Please describe any incentives (e.g. resending invitations to participate or reimbursing participants for their time) used to encourage participation
- Please outline what language(s) the Delphi was conducted in and any adaptations to make this more accessible (e.g. translations or plain language summaries)
3. Results
- Results are clear and well-reported
4. Discussion
- The discussion section is relatively short and there are minimal links between existing research and your results. Please expand on this, including more detail on if the recommendations you created are consistent with the existing literature and if not potential reasons for this.
- Please include more information on potential limitations.
5. References
- Appropriate references are used and your referencing style is sound.
6. Other issues
- Please check for typos, e.g. on page 7 line 250 there seems to be an extra 0 next to reference 23
- If possible, please include copy of focus group guides/surveys in supplementary material
I look forward to reading the next version of the paper!
Round 2
Reviewer 3 Report
Comments and Suggestions for Authors
Thank you for the opportunity to review the revised version of this paper. All of comments have been appropriately addressed and I now feel the paper is suitable for publication.